# Establishment of the experimental procedure for prediction of conjugation capacity in mutant UGT1A1

Yutaka Takaoka[1]*, Atsuko Takeuchi[2], Aki Sugano[1], Kenji Miura[1], Mika Ohta[1], Takashi Suzuki[1], Daisuke Kobayashi[3], Takuji Kimura[4], Juichi Sato[4], Nobutaro Ban[4,5], Hisahide Nishio[6], Toshiyuki Sakaeda[7]

1 Division of Medical Informatics and Bioinformatics, Kobe University Hospital, Kobe, Japan, 2 Division of Analytical Laboratory, Kobe Pharmaceutical University, Kobe, Japan, 3 Division of Medical and Healthcare Systems, Healthcare Economics and Hospital Administration, Kobe University Graduate School of Medicine, Kobe, Japan, 4 Department of General Medicine/Family and Community Medicine, Nagoya University Graduate School of Medicine, Nagoya, Japan, 5 Medical Education Center, Aichi Medical University School of Medicine, Nagakute, Aichi, Japan, 6 Department of Community Medicine and Social Healthcare Science, Kobe University Graduate School of Medicine, Kobe, Japan, 7 Department of Pharmacokinetics, Kyoto Pharmaceutical University, Kyoto, Japan

* ytakaoka@ebraille.med.kobe-u.ac.jp

**Data Availability Statement:** All relevant data are within the paper, its Supporting Information files, and from figshare at https://doi.org/10.6084/m9.figshare.10252772.v1.

## Abstract

UDP-glucuronosyltransferase 1A1 (UGT1A1) is an enzyme that is found in the endoplasmic reticulum membrane and can reportedly have a large number of amino acid substitutions that result in the reduction of glucuronidation capacity. For example, adverse drug reactions when patients receive CPT-11 (irinotecan) such as in cancer chemotherapy are caused by amino acid substitutions in UGT1A1. We previously found that the extent of the docking when the hydroxyl residue of bilirubin was oriented toward UDP-glucuronic acid correlated with *in vitro* conjugation capacity. In this study, we analyzed the conformation of mutant UGT1A1s by means of structural optimization with water and lipid bilayers instead of the optimization *in vacuo* that we used in our previous study. We then derived a mathematical model that can predict the conjugation capacities of mutant UGT1A1s by using results of substrate docking *in silico* and results of *in vitro* analysis of glucuronidation of acetaminophen and 17β-estradiol by UGT1A1s. This experimental procedure showed that the *in silico* conjugation capacities of other mutant UGT1A1s with bilirubin or SN-38 were similar to reported *in vitro* conjugation capacities. Our results suggest that this experimental procedure described herein can correctly predict the conjugation capacities of mutant UGT1A1s and any substrate.

## Introduction

Uridine diphosphate glucuronosyltransferase 1A1 (UGT1A1) is a member of the UDP-glucuronosyltransferase 1A enzyme family that is mainly localized in the smooth endoplasmic reticulum in the liver and other tissues [1]. UGT1A1 plays an essential role in the metabolism of

**Funding:** This research was partially supported by Grants-in-Aid from Japanese Ministry of Education, Culture, Sports, Science and Technology (https://www.jsps.go.jp/english/index.html) No. 18K07414 (to Y.T.), No. 19K12202 (to A.S.), and No. 19K07867 (to M.O.), and Grants-in-Aid from Hyogo Science and Technology Association (http://hyogosta.jp) No. 27004 (to Y.T.). The funders had no role in study design, data collection and analysis, decision to publish, or preparation of the manuscript.

**Competing interests:** The authors have declared that no other competing interests exist than stated in financial disclosure.

about 80 chemical substances including bilirubin and CPT-11 (irinotecan), an anticancer agent [2, 3]. Locuson and Tracy [4] reported on a binding model of UGT1A1 with a coenzyme, UDP-glucuronic acid (UDPGA), whose binding site included the amino acids S38, H173, G308, L355, S375, H376, and G377. In our previous research based on their report, we showed that conjugation proceeded when the hydroxyl residue of bilirubin was oriented toward UDPGA [5].

Mutations of the *UGT1A1* gene have been found in hereditary diseases, such as hyperbilirubinemia, Crigler-Najjar syndrome, and Gilbert syndrome, and at least 70 mutations of this gene have been reported [6]. An adverse reaction to the drug CPT-11 is well known to be associated with mutant *UGT1A1* [7]. Major mutations of *UGT1A1*, especially the *UGT1A1*6* and *UGT1A1*28* mutations, are well known to reduce conjugation capacity. The former mutation causes the loss of function in conjugation capacity because of the amino acid substitution of G71R [8], and the latter induces a quantitative reduction in *UGT1A1* gene expression with an increase in the TA repeat from 6 to 7 of the promoter region [9, 10]. CPT-11 is converted to SN-38, an active metabolite, and is then metabolized to become a water-soluble form via glucuronidation of the UGT1A1 enzyme [11]. The risk for CPT-11 toxicity increases with these genetic variants. Today, a genetic test for *UGT1A1*, especially the *UGT1A1*6* and *UGT1A1*28* mutations, is performed before treatment with CPT-11, because use of CPT-11 carries a high risk of adverse effects such as severe neutropenia and diarrhea [12]. In addition, sequencing technology can determine the new amino acid substitutions in UGT1A1, but the glucuronidation capacities of these new mutant UGT1A1s for any substrate, such as SN-38 are usually unknown.

Our molecular simulation analysis that was based on structural biology recently became a powerful tool for studying the biochemical process [13–18]. Previously, our *in silico* study of the conjugation process showed that this enzymatic reaction was controlled by the direction of the vicinal hydroxyl group in the substrate [5]. In addition, mathematical models are used for research on biological processes, such as those in drug-drug interactions (DDIs) and drug metabolism [19, 20].

For our research reported here, we revised our previous molecular simulation analyses, which led to a mathematical model for the conjugation process in order to predict the conjugation capacity for substrates such as bilirubin and SN-38. Coenzyme or substrate docking after structural optimization with water and lipid bilayers was performed to mimic the cellular environment instead of the *in vacuo* environment that we had used in our previous research. We subsequently developed a mathematical model that we derived from the results from *in vitro* and *in silico* analyses. Then we demonstrate here that our mathematical model using the docking results for a specific number of UDPGA-oriented hydroxyl residues of substrates can predict the conjugation capacities of mutant UGT1A1s.

## Materials and methods

### Construction of the 3D structures of UGT1A1 mutants

The 3D structure of wild-type UGT1A1 was obtained from ModBase [21] (Model ID: 2420a568b0f3d1b1fe06fc34a94eee40). We then added hydrogen atoms to the model structure via PyMOL software [22], after which we prepared structures of UGT1A1 mutants: G71R, F83L, P229L, P229Q, L233R, I294T, I322V, R336L, H376R, P387S, N400D, and W461R, whose PMIDs in the Protein Model Database [23] are PM0082268, PM0082269, PM0082278, PM0082271, PM0082279, PM0082275, PM0082270, PM0082272, PM0082273, PM0082274, PM0082276, and PM0082277, respectively. We prepared these structures by using molecular operating environment (MOE) software (Chemical Computing Group Inc., Montreal, QC, Canada). Each model structure of the wild-type UGT1A1 and mutant UGT1A1s was

embedded in the lipid bilayer using visual molecular dynamics (VMD) software [24]. The location of the bilayer was determined according to a previous report [25]. The transferable intermolecular potential with three points (TIP3P) water model [26] was used for structural optimization of UGT1A1 with water. The minimal distance of a protein atom to the edge of the rectangular water box was 14 Å.

The structural data were then subjected to structural optimization with the AMBER10:EHT force field according to our previous study [27] with slight modifications. We performed energy minimization by means of the molecular mechanics (MM) function of the MOE software by using the steepest descent method until the root mean square gradient was 0.01 kcal/mol/Å. Then, we used NAMD software to perform molecular dynamics (MD) simulations [28]. The water-protein-bilayer system was gradually heated from 0 K to 310 K during 250 ps. After the heating process, a 10,000-ps production run was performed with the NPT ensemble in a unit of 2 fs. The temperature and pressure of the system were maintained by using the Berendsen coupling algorithm [29]. The SHAKE algorithm [30] was used to constrain water bond geometries. The root mean square deviation (RMSD) from the 10,000-ps molecular dynamics trajectory was analyzed by using VMD (S1A Fig). Each trajectory was stable after 8000 ps. The quality of each 3D UGT1A1 structure was ascertained via the PROCHECK program [31], which yielded 1.8% or less in the disallowed regions of the Ramachandran plot (S1B and S1C Fig). This provided an insight into the correctness of the modeled structures in terms of PROCHECK as reported previously [32]. Each UGT1A1 molecule in the last frame (S2 Fig) was used for the docking analysis.

## Docking analysis of UGT1A1 with UDPGA

The docking of UDPGA and each UGT1A1 mutant was analyzed by means of AutoDock4 [33] according to our previous research [5] with slight modifications. The docking site was defined by using the AutoGrid program as a box within 3.8 Å of the amino acids that reportedly interact with UDPGA [4]. This value was determined by means of the sum of the H bond distance (3.2 Å) [34] and the error of the AutoDock (0.6 Å), which we found in our preliminary analysis (S1 Text, S1 Table). Grids were searched via the Lamarckian genetic algorithm. All other parameters were defined by default settings. All hydrogen atoms were added and water molecules were removed. One hundred different docking runs were performed for each UGT1A1-UDPGA pair.

Docking for each UGT1A1 molecule and the coenzyme UDPGA was performed in the binding mode similar (wild-type) or neighboring (mutants) to that in previous research [4], which are comprised of amino acids S38, H173, G308, L355, S375, H376, G377, and was defined as the correct binding mode.

## Analysis of UGT1A1 docking with acetaminophen (AAP) and 17β-estradiol (E2)

We simulated AAP and E2 docking to each UGT1A1 mutant by means of the AutoDock4 program. The 3D structural data of the substrates were obtained from ChemIDplus (registry numbers: 103-90-2 for AAP, 50-28-2 for E2).

The docking site for each of the correctly bound complexes of UGT1A1 and UDPGA was located in front of the bound UDPGA and was sufficiently large to cover the binding site for all substrates that we used in this study. The far boundary of the docking site was defined as 0.5 Å back of the center of the C-O bond between glucuronic acid and UDP of UDPGA.

We performed 100 docking runs for each UGT1A1-substrate pair. We analyzed the number of docking poses with a hydroxyl group oriented toward the coenzyme, that is, the hydroxyl

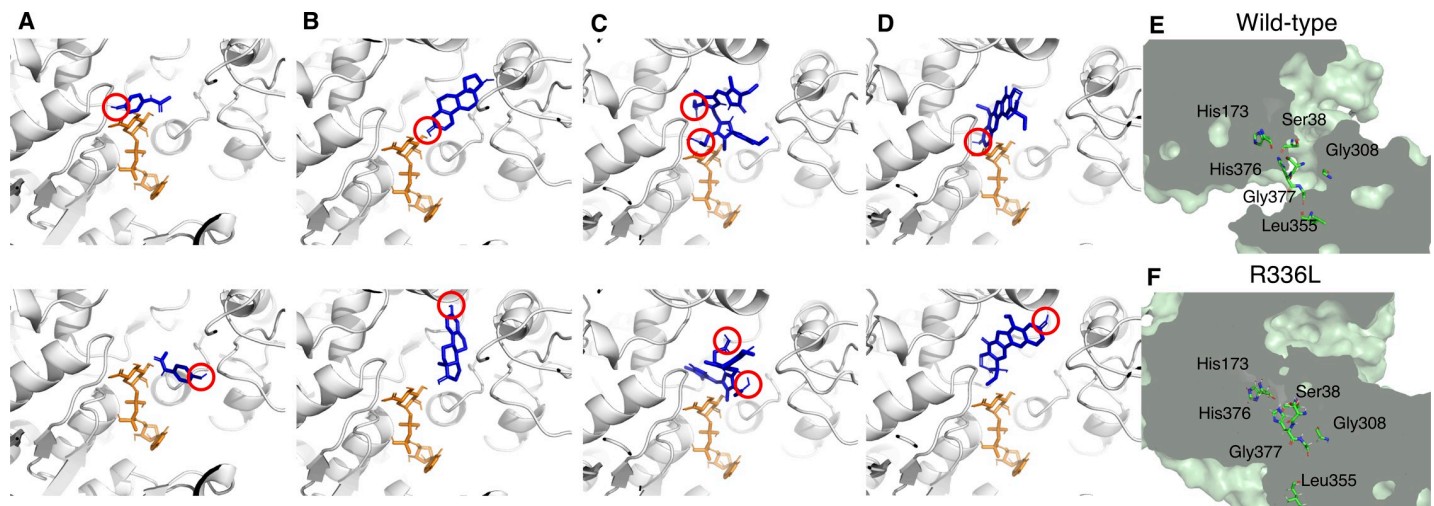

**Fig 1. The complexes of UGT1A1 with coenzyme and substrates.** Pairs of UDPGA and substrates for wild-type UGT1A1 are as follows: A: UDPGA and AAP; B: UDPGA and E2; C: UDPGA and bilirubin; and D: UDPGA and SN-38. Top panels in each pair show the binding modes that can function in the conjugation reaction: the hydroxyl group of the substrate (red circle) is oriented toward UDPGA, which we call hydroxyl orientations; bottom panels show the binding modes that cannot function in the conjugation reaction. The UGT1A1 structures are shown as ribbon representations. UDPGA and substrates are shown as orange and blue sticks, respectively. (E and F) Cross-sectional images of wild-type UGT1A1 and R336L UGT1A1 structures highlighting the location of the glucuronidation and the amino acids of the coenzyme binding. The field of the conjugation process was disrupted in R336L.

group of the substrate was located within 3.8 Å of the center of the C-O bond between glucuronic acid and UDP of UDPGA, which we had identified as the hydroxyl orientation and the correct binding mode in our previous report [5]. Fig 1A, 1B, 1C and 1D show representative docking results of hydroxyl orientations and other orientations that cannot function in the conjugation reaction. Fig 1E shows the field of the conjugation process in wild-type but the field completely lacks in R336L mutant (Fig 1F).

### *In vitro* analysis of glucuronidation for AAP and E2

*In vitro* conjugation capacities of wild-type UGT1A1 and mutant UGT1A1s were analyzed as described previously [35] with a slight modification. Human UGT1A1 cDNA was obtained from a human liver cDNA library by PCR amplification and was inserted into a pENTR/ D-TOPO vector. Mutations were introduced by using a site-directed mutagenesis method with substitutions at nucleotide positions c.211G>A (p.G71R), c.247T>C (p.F83L), c.964A>G (p.I322V), c.1007G>T (p.R336L), c.1127A>G (p.H376R), and c.1159C>T (p. P387S).

Each wild-type and mutant UGT1A1 sequence was inserted into the expression vector pcDNA-DEST40 Gateway by recombination. The UGT1A1 expression vectors, together with a luciferase reporter vector (pGL3 vector) as an indicator of transfection efficiency, were transfected into COS-7 cells by using Lipofectamine 2000, according to the manufacturer's instructions. The cells were harvested at 48 h after transfection and then homogenized with 70 μL of 0.1 M Tris–HCl (pH 7.5). The cell homogenates were used for UGT1A1 activity and luciferase assays. Luciferase activity was measured with a TD-20/20 luminometer (Promega, Madison, WI, USA) and was used to normalize enzyme activities of the UGT1A1 clones.

Glucuronidation of AAP and E2 was analyzed with the UGT Reaction Mix containing UDPGA (Becton, Dickinson and Company, Franklin Lakes, NJ, USA), according to the manufacturer's instructions. Briefly, a 99-μL sample of the reaction mixture containing 20 μL of Solution A, 8 μL of Solution B, 1 μL of 20 mM AAP or 0.25 mM E2 in ethanol, and 70 μL of

cell homogenates was incubated at 37°C for 2 h. The reaction was terminated by adding 25 μL of acetonitrile for AAP or 25 μL of 94% acetonitrile/6% glacial acetic acid for E2.

After the reaction termination, 5 μL of 1mM *p*-nitrophenyl β-D-glucuronide (PNPG) or 1 μL of 300 μM ethynylestradiol (EE2) in ethanol was added to the reaction mixture as an internal standard. Then, the reaction mixture was centrifuged and the supernatant was subjected to liquid chromatography-tandem mass spectrometry (LC-MS/MS) to determine the amount of AAP glucuronide (AAPG) or E2 glucuronide (E2G).

Chromatography was performed with a Shimadzu LC10AD system (Shimadzu, Kyoto, Japan) with a mobile phase consisting of 40% acetonitrile and 25 mM ammonium acetate at a flow rate of 0.3 mL/min for AAP, or 70% methanol, 4.5% of acetonitrile, and 0.15 mM perchloric acid at a flow rate of 0.2 mL/min for E2. The column temperature was maintained at 45°C. A 15-μL aliquot of each sample was injected onto a Shim-pack CLC-ODS column (4.6 mm I. D. × 15 cm; Shimadzu).

An API-3000TM LC-MS/MS system (Applied Biosystems/MDS SCIEX, Toronto, Canada) was operated with an electrospray ionization source (ESI; TurboIonSpray interface) coupled with the LC system described above. MS scanning was operated in a negative ion mode. Analyst 1.3.1 software was used for equipment control, data acquisition, and analysis. The glucuronide peaks were detected at *m/z* 326.1/150.0 [precursor ion (Q1)/product ion (Q3)] for AAPG and *m/z* 447.1/271.1 (Q1/Q3) for E2G. These glucuronide peaks were confirmed because their retention times were the same as those of commercially available authentic standards. With regard to the sensitivity of all assays, the minimal level of detection was 3 ng/assay. Experiments were repeated five times; the coefficient of variation was less than 12%. S3 Fig shows representative LC-MS/MS chromatograms.

## Mathematical model for conjugation capacity of UGT1A1

To establish a mathematical model for glucuronidation by UGT1A1, the enzymatic reaction of UGT1A1 was described by an equation as follows: we first compared *in vitro* conjugation capacity with the results of docking simulation analyses; we then derived an equation that calculated *in vitro* conjugation capacity by using the docking results that showed a strong correlation with *in vitro* conjugation capacity. Different efficiencies of glucuronidation with individual substrates were incorporated into the equation by using the docking results for wild-type UGT1A1.

## Verification of the mathematical model by cross-validation

To verify the mathematical model, we used leave-one-out cross-validation with AAP and E2 as substrates as follows: We removed one of the UGT1A1 mutants, determined the constants of the mathematical model by using the rest of the mutants, and then performed calculations to predict the *in silico* conjugation capacity for the excluded mutant. Finally, we calculated Pearson's correlation coefficient for the *in silico* and the *in vitro* conjugation capacities.

## Validation of the experimental procedure: Prediction of conjugation capacity of bilirubin and SN-38

To examine the validity of this experimental procedure, we prepared structures of the mutants P229Q, I294T, N400D, W461R, P229L, and L233R for only bilirubin and/or SN-38. The 3D structures of bilirubin and SN-38 were obtained from ChemIDPlus (registry numbers: bilirubin, 635-65-4; SN-38, 86639-52-3) for docking analyses. The docking analyses were performed for the pairs of substrates and UGT1A1 mutants whose *in vitro* conjugation capacities were reported. To use our mathematical model, docking analyses of bilirubin with UGT1A1s (wild-

type, G71R, F83L, P229Q, I294T, N400D, W461R) and SN-38 with UGT1A1s (wild-type, G71R, P229L, P229Q, L233R) were performed in the same manner as those for AAP and E2. The *in silico* conjugation capacity of each substrate was calculated by means of our mathematical model and then compared with the reported *in vitro* conjugation capacity. Finally, a correlation plot was analyzed for verification of the mathematical model by using all prediction results for *in silico* and *in vitro* data.

### Statistical analysis

Statistical analyses were performed with R software (R Core Team 2015). Data are presented as means ± SD. A *P* value of <0.05 was regarded as statistically significant.

## Results

### Correlation between the results of docking simulation and *in vitro* conjugation capacity

To evaluate the results of molecular simulation analyses for the mathematical model for glucuronidation by UGT1A1, we compared the *in vitro* conjugation capacity of UGT1A1 against AAP and E2 with the results of docking simulation analysis (Fig 2, S2 Table). A significant correlation existed between the number of hydroxyl orientations of substrate and *in vitro* conjugation capacity, as we reported previously [5] (Table 1).

A correlation also existed between the number of hydroxyl orientations of the substrate and *in vitro* conjugation capacity for the other UGT1A isoforms, UGT1A10 and UGT1A7 (S2 Text, S4 Fig). However, no significant correlation was found between the binding mode of UDPGA and *in vitro* conjugation capacity. These results suggest that the hydroxyl orientations of the substrate can be used to derive the mathematical model for glucuronidation by UGT1A1.

### Derivation of a mathematical model for estimating the conjugation capacity of UGT1A1

A mathematical model to predict the conjugation capacity of UGT1A1 was derived as described below. Glucuronidation involves the following steps: (i) binding of coenzyme UDPGA to UGT1A1; (ii) binding of substrate to the UGT1A1-UDPGA complex; (iii) conjugation of glucuronic acid with the substrate [36]. The conjugation capacity of UGT1A1 ($P_C$) can be represented as the product of (i) and (ii):

$$P_c = C \times S \tag{1}$$

where $C$ is the contribution of (i) to conjugation capacity and $S$ is the contribution of (ii) to conjugation capacity.

The difference in the docking results for each substrate with wild-type UGT1A1 (Fig 2B and 2C) is represented by the substrate-specific constant σ, and ε represents the *in vivo* environment of the enzymatic reaction::

$$P_c = \sigma \times C \times S + \varepsilon \tag{2}$$

The value of ε is set to 0 to predict *in vitro* conjugation capacity.

Because no correlation exists between the correct binding mode of UDPGA and *in vitro* conjugation capacity (Table 1), $C$ is replaced with 1:

$$P_c = \sigma \times S + \varepsilon \tag{3}$$

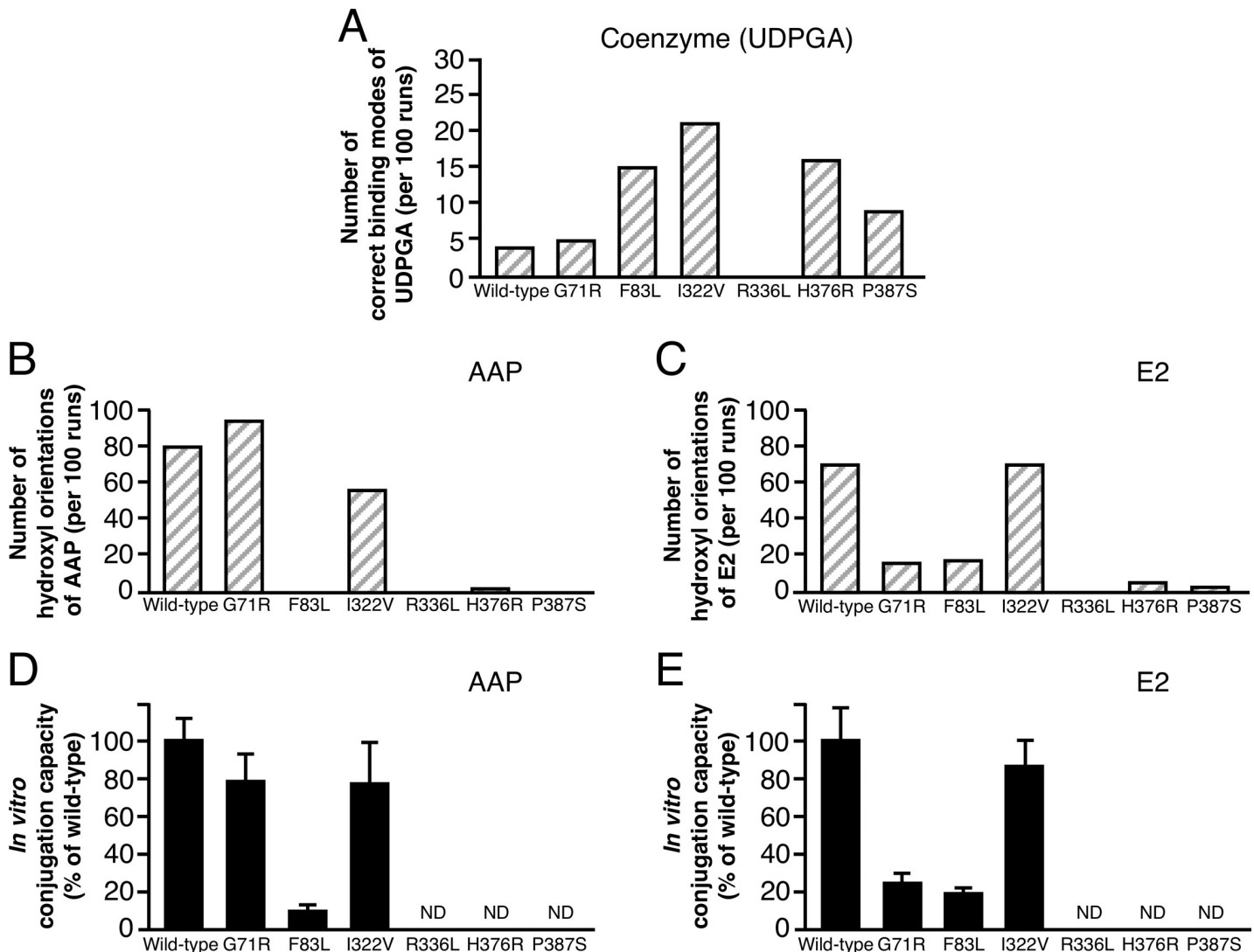

**Fig 2. Comparison between docking results and *in vitro* conjugation capacities of wild-type UGT1A1 and UGT1A1 mutants for AAP and E2.** (A) The number of correct binding modes of UDPGA per 100 separate docking runs. (B and C) The number of hydroxyl orientations of substrate per 100 separate docking runs. S2 Table provides raw data for docking results for panels A, B, C, and D. (D and E) *In vitro* conjugation capacity. ND, not detected.

$S$ is defined as

$$S = \left(\frac{S_{DH}}{S_{DT}}\right)^{\beta_l} \ where \ \beta_l = \frac{S_{DT}}{S_{DH,wild}} \tag{4}$$

where $S_{DT}$ is the total number of substrate docking and $S_{DH}$ is the number of hydroxyl orientations of the substrate [5].

**Table 1. Correlation of docking simulation results with *in vitro* conjugation capacity of UGT1A1.**

| Substrate | Correct binding mode of UDPGA | | Hydroxyl orientation of substrate | |
|---|---|---|---|---|
| | Correlation coefficient | *P* value | Correlation coefficient | *P* value |
| AAP | - 0.189 | 0.7195, >0.05 | 0.954 | 0.0032, <0.01 |
| E2 | 0.114 | 0.8298, >0.05 | 0.994 | 0.00005, <0.001 |

Because the enzymatic reaction follows a sigmoid curve [37], $S$ can be defined by using a sigmoid function:

$$S = \left( \frac{1}{1 + e^{-\gamma_l \left( \frac{S_{DH}}{S_{DT}} - \mu_l \right)}} \right)^{\beta_l} \text{ where } \beta_l = \frac{S_{DT}}{S_{DH,wild}} \tag{5}$$

In addition, we define a variable $\kappa$ to represent TA repeat polymorphism (UGT1A1*28) in the promoter region that is based on the relative values to wild-type/wild-type for each genotype according to the $V_{\max}$ values in the previous report: wild-type/wild-type, 16.2 nmol/min/mg; wild-type/*28, 12.0 nmol/min/mg; *28/*28, 3.4 nmol/min/mg [10]:

$$P_c = \sigma \times \kappa \times \left( \frac{1}{1 + e^{-\gamma_l \left( \frac{S_{DH}}{S_{DT}} - \mu_l \right)}} \right)^{\beta_l} + \varepsilon \tag{6}$$

$$\text{where } \kappa = \begin{cases} 1.0 & \text{wild−type/wild−type} \\ 0.74 & \text{wild−type/*28} \\ 0.21 & \text{*28/*28} \end{cases}$$

Constant values $\sigma$, $\gamma_l$, $\mu_l$, and $\varepsilon$ were estimated by minimizing the sum of squared error between *in silico* conjugation capacity $P_C$ and *in vitro* conjugation capacity $V_c$:

$$(\sigma, \gamma_l, \mu_l, \varepsilon) = argmin \left\{ \sum_M (P_c - V_c)^2 \right\} \tag{7}$$

where set $M$ represents the UGT1A1 mutants whose *in vitro* conjugation capacities are known and correct binding modes of UDPGA were obtained by docking analysis.

The relative conjugation capacity of mutant UGT1A1 (percentage of wild-type) is defined by following equation:

$$P_c = \frac{\sigma \times \kappa \times \left( \frac{1}{1 + e^{-\gamma_l \left( \frac{S_{DH}}{S_{DT}} - \mu_l \right)}} \right)^{\beta_l} + \varepsilon}{\sigma \times \left( \frac{1}{1 + e^{-\gamma_l \left( \frac{S_{DH,wild}}{S_{DT}} - \mu_l \right)}} \right)^{\beta_l} + \varepsilon} \times 100 \tag{8}$$

In the case of $\varepsilon = 0$, Eq (8) is represented by Eq (9):

$$P_c = \frac{\kappa \times \left( \frac{1}{1 + e^{-\gamma_l \left( \frac{S_{DH}}{S_{DT}} - \mu_l \right)}} \right)^{\beta_l}}{\left( \frac{1}{1 + e^{-\gamma_l \left( \frac{S_{DH,wild}}{S_{DT}} - \mu_l \right)}} \right)^{\beta_l}} \times 100 \tag{9}$$

S2 Table shows the variables and constants of the mathematical model for the docking results for each UGT1A1 variant.

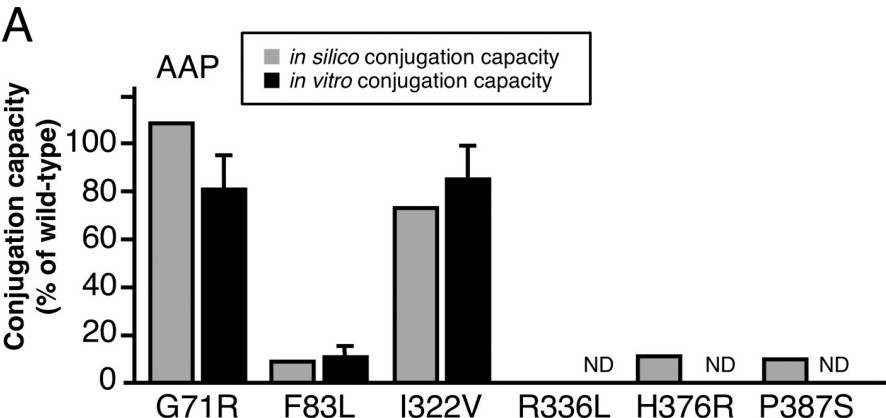

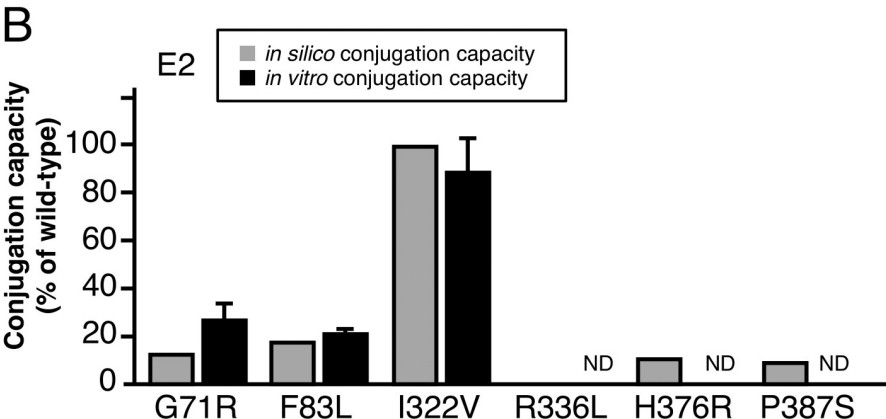

| Substrate | Correlation coefficient | *P* value |
|---|---|---|
| AAP | 0.973 | 0.0111, <0.05 |
| E2 | 0.981 | 0.0078, <0.01 |

**Fig 3. Cross-validation of *in silico* conjugation capacity of UGT1A1 with *in vitro* conjugation capacity.** (A) *In silico* conjugation capacities (gray bars) of AAP were compared with *in vitro* conjugation capacities (black bars). (B) *In silico* and *in vitro* conjugation capacities of E2 were compared, as done for AAP. (C) Pearson's correlation coefficient was used to assess cross-validation. ND, not detected.

## Validation of the mathematical model

To verify our mathematical model, we validated Eq (9) by using leave-one-out cross-validation and leaving out one mutant per fold. We calculated the *in silico* conjugation capacity of AAP and E2 by using the docking results (S2 Table), and we then compared the result with *in vitro* conjugation capacity (Fig 3A and 3B). We found a significant correlation between *in silico* conjugation capacity and *in vitro* conjugation capacity (Fig 3C). On the basis of these results, we confirmed our mathematical model to be useful for predicting the conjugation capacity of mutant UGT1A1.

## Validation of the experimental procedure: Prediction of the conjugation capacity of bilirubin and SN-38

We performed the docking analyses of UGT1A1 with bilirubin and SN-38 in the same manner as those for AAP and E2 (S2 Table). The *in silico* conjugation capacities were calculated by inserting the docking results into Eq (9), and we then compared these results with the reported *in vitro* conjugation capacities [8, 38–41] (Figs 4, 5A and 5B). We found significant correlations between *in silico* and *in vitro* conjugation capacities (Fig 5C). These results indicate that our method for the prediction of conjugation capacity is valid in the tested cases of bilirubin and SN-38. Fig 6 indicates that our experimental procedure is effective for predicting the conjugation capacity, because all *in silico* results and *in vitro* results were significantly correlated.

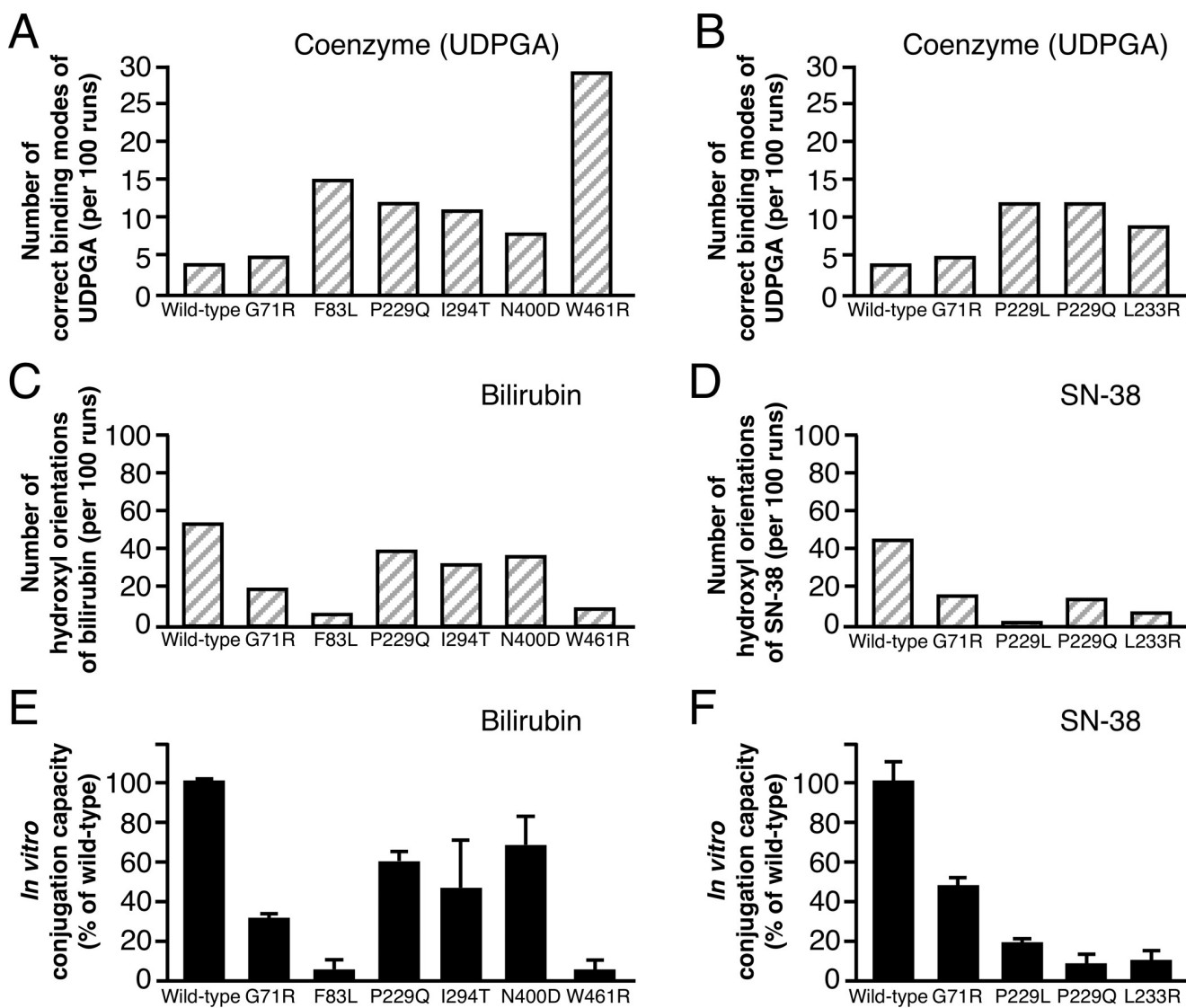

**Fig 4. Comparison between docking results and *in vitro* conjugation capacities of wild-type UGT1A1 and UGT1A1 mutants for bilirubin and SN-38.** (A) The number of correct binding modes of UDPGA per 100 separate docking runs. (B and C) The number of hydroxyl orientations of substrate per 100 separate docking runs. S2 Table provides raw data for docking results for panels A, B, C, and D. (E and F) *In vitro* conjugation capacity.

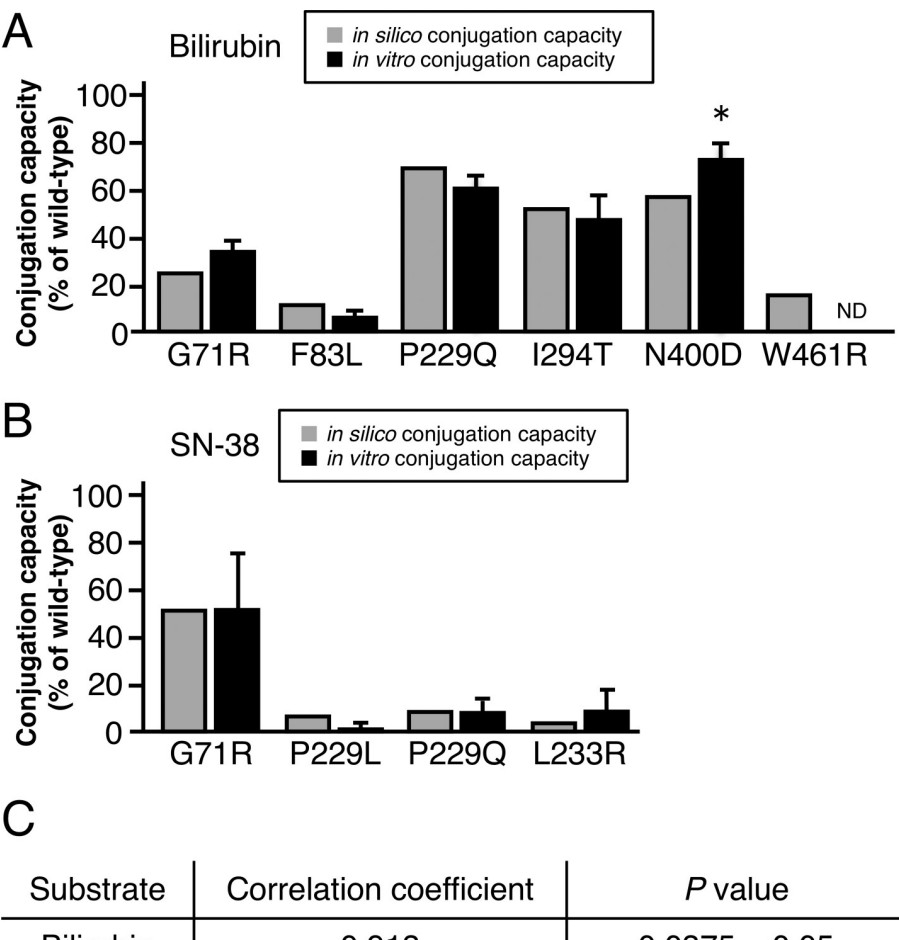

**Fig 5. Verification of the prediction of *in silico* conjugation capacities of UGT1A1 mutants.** (A) *In silico* conjugation capacities (gray bars) for bilirubin were calculated by using molecular docking results and our mathematical model and were compared with reported *in vitro* conjugation capacities (black bars). (B) *In silico* and *in vitro* conjugation capacities of SN-38 were compared similarly, as done for bilirubin. (C) Pearson's correlation coefficient was used to analyze the correlation between *in silico* and *in vitro* conjugation capacities. *, Calculated from *in vivo* conjugation capacity [5]. ND, not detected.

## Discussion

Adverse drug reactions and differences in drug efficacies are often caused by mutations in drug metabolizing agents, drug targets, and drug transporters [42]. Recently, novel mutations causing amino acid substitutions have been found by genome analysis with high-throughput sequence technology [43]. These mutations cause quantitative reductions and loss of function in the enzymes [44]. Drug efficacy differences and adverse drug reactions generally do not occur without major mutations and polymorphism in promoter regions. If novel mutants are found, genetic information is unknown, and therefore genetic analysis is not used in a clinical therapeutic setting.

In this research, we developed *in silico* procedures that consists of a mathematical model and molecular simulation analyses, and we evaluated its validity. We strictly selected and specified software for molecular simulation and parameter settings to mimic the conjugation

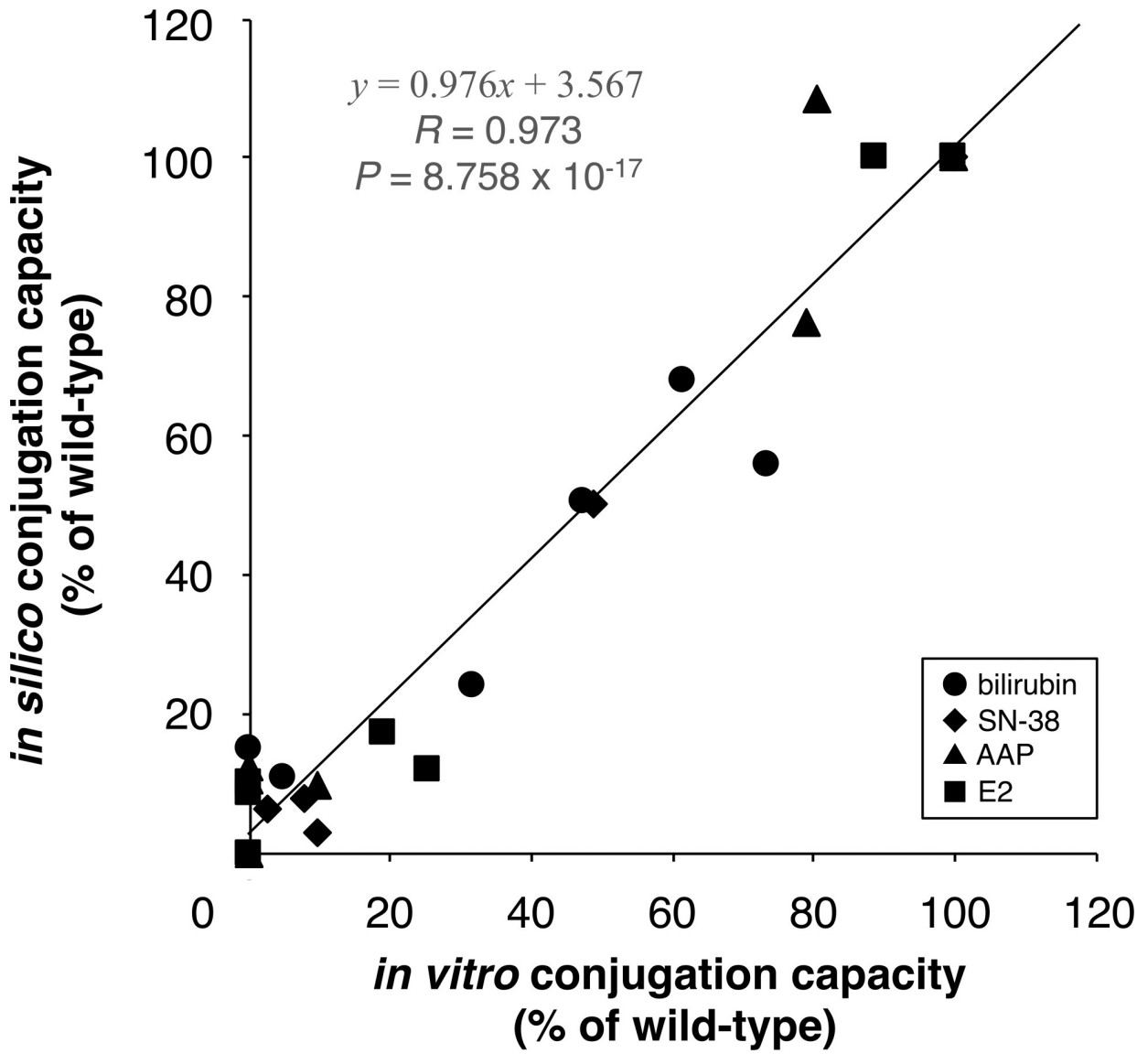

**Fig 6. Correlation analysis between *in silico* and *in vitro* conjugation capacities.** Regression equation (solid line), Pearson's correlation coefficient (R), and P value are shown.

process by UGT1A1, and the correctness of all modeled structures in this research was confirmed by the stable RMSD trajectory and low percentages in disallowed regions of Ramachandran plots.

To derive our mathematical model, we both performed docking analyses of UGT1A1 with the coenzyme or substrates (AAP and E2) and obtained the *in vitro* conjugation capacities of G71R, F83L. I322V, R336L, H376R, and P387S mutant of UGT1A1s. We compared *in silico* and *in vitro* results for two substrates to identify the key factors in this enzymatic reaction. We then found the number of hydroxyl orientations of substrate obtained from docking simulation analysis, which we identified as one feature (Fig 2, Table 1) that we previously identified as being a key assumption [5]. In our docking analysis using the AutoDock program, similar docking results were acquired by the docking program of MOE software. Moreover, we found the same feature with other UGT1A enzymes: UGT1A7 and UGT1A10 (S4 Fig). These results

suggest that the conjugation capacities of UGT1As are regulated by the number of hydroxyl orientations of the substrates. After our comparative analysis, we derived the mathematical model including the number of docking poses with correct binding, and it correctly determined the conjugation capacity of UGT1A1 mutants when conjugated with AAP and E2 (Fig 3).

Finally, our procedure *in silico*, which consists of molecular simulation analyses and our mathematical model, was validated by its use for the glucuronidation of bilirubin and SN-38. We compared the calculated conjugation capacities *in silico* with reported conjugation capacities *in vitro* [8, 38–41]. These predicted conjugation capacities closely approximated the reported *in vitro* findings (Fig 5). These results indicated that our mathematical model is valuable to predict the capacity of UGT1A1 to metabolize drugs. In this research, our procedure demonstrates the potential to predict the capacity of all UGT1A1 mutants based on Fig 6, which shows the data of the most studied mutants *in vitro*, and it also suggests that the same strategy may be available for other metabolic enzymes. This experimental procedure thus provides important information about adverse drug reactions for clinicians.

## Supporting information

**S1 Fig. Validation of the 3D structures of wild-type UGT1A1 and UGT1A1 mutants.** (A) The RMSD curves of the 10,000-ps trajectories for the backbone atoms of the UGT1A1 mutants were calculated with respect to the initial structures as a function of time. (B) Ramachandran plots for UGT1A1s. (C) Plot statistics for each UGT1A1.
(TIFF)

**S2 Fig. Cartoon representations of 3D structures of wild-type UGT1A1 and mutant UGT1A1s that we used in this study.**
(TIFF)

**S3 Fig. LC-MS/MS analysis of AAPG and E2G.** Representative chromatograms of (A) AAPG and (B) E2G in the reaction mixtures include authentic standard AAPG and E2G, reaction mixtures without UGT1A1, and mixtures with wild-type UGT1A1 or G71R-mutant UGT1A1. AAPG and E2G were detected in the reaction mixtures with wild-type UGT1A1 and G71R-mutant UGT1A1, whereas glucuronides were absent in the reaction mixtures without UGT1A1.
(TIFF)

**S4 Fig. Comparison between docking results and reported *in vitro* conjugation capacities of UGT1A7 and UGT1A10.** (A and B) The number of correct binding modes of UDPGA per 100 separate docking runs. (C and D) The number of correct binding modes (hydroxyl orientations) of substrate per 100 separate docking runs. (E and F) Reported *in vitro* conjugation capacity. A correlation was shown between the hydroxyl orientation of the substrate and the *in vitro* conjugation capacity.
(TIFF)

**S1 Table. Intermolecular hydrogen bond distance of VvGT and UDP in the crystal structure and the docking results.**
(DOCX)

**S2 Table. Docking results for UGT1A1 with UDPGA, AAP, E2, bilirubin, and SN-38.**
(DOCX)

**S3 Table. Variables and constants of the mathematical model.**
(DOCX)

**S1 Text. Analysis of the conformational difference between the docking model and the crystal structure.**
(DOCX)

**S2 Text. Correlation between docking simulation results and *in vitro* conjugation capacity of UGT1A7 and UGT1A10.**
(DOCX)

## Author Contributions

**Conceptualization:** Yutaka Takaoka, Hisahide Nishio, Toshiyuki Sakaeda.

**Data curation:** Yutaka Takaoka, Aki Sugano.

**Formal analysis:** Yutaka Takaoka, Aki Sugano, Daisuke Kobayashi.

**Funding acquisition:** Yutaka Takaoka, Aki Sugano, Mika Ohta.

**Investigation:** Yutaka Takaoka, Atsuko Takeuchi, Aki Sugano, Kenji Miura, Mika Ohta, Takuji Kimura.

**Methodology:** Yutaka Takaoka, Takashi Suzuki, Juichi Sato, Nobutaro Ban, Hisahide Nishio, Toshiyuki Sakaeda.

**Project administration:** Yutaka Takaoka.

**Resources:** Yutaka Takaoka, Atsuko Takeuchi, Aki Sugano.

**Software:** Yutaka Takaoka, Aki Sugano.

**Supervision:** Yutaka Takaoka.

**Validation:** Yutaka Takaoka, Aki Sugano, Takashi Suzuki, Juichi Sato, Nobutaro Ban, Hisahide Nishio, Toshiyuki Sakaeda.

**Visualization:** Yutaka Takaoka, Aki Sugano.

**Writing – original draft:** Yutaka Takaoka, Atsuko Takeuchi, Aki Sugano, Kenji Miura.

**Writing – review & editing:** Yutaka Takaoka, Aki Sugano, Nobutaro Ban.

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
