## [Decision Letter · Decision Letter 0]

17 Sep 2019

PONE-D-19-22725

Establishment of the experimental procedure for prediction of conjugation capacity in mutant UGT1A1

PLOS ONE

Dear Dr. Takaoka,

Thank you for submitting your manuscript to PLOS ONE. After careful consideration, we feel that it has merit but does not fully meet PLOS ONE’s publication criteria as it currently stands. Therefore, we invite you to submit a revised version of the manuscript that addresses the points raised during the review process.

We would appreciate receiving your revised manuscript by Nov 01 2019 11:59PM. To enhance the reproducibility of your results, we recommend that if applicable you deposit your laboratory protocols in protocols.io, where a protocol can be assigned its own identifier (DOI) such that it can be cited independently in the future. For instructions see: http://journals.plos.org/plosone/s/submission-guidelines#loc-laboratory-protocols

We look forward to receiving your revised manuscript.

Kind regards,

Douglas Dluzen

Academic Editor

PLOS ONE

Journal Requirements:

Additional Editor Comments:

Dear Dr. Takaoka,

Thank you for submitting to PLOS ONE. After careful review, I have decided that a major revision is required for your manuscript. You will find each reviewer's comments attached with this decision letter. It would be helpful to have all of these comments addressed if you decide to revise. As well, I am in strong agreement with Reviewer 1 that M/S traces are provided as supplementary for your data, including a new primary figure with example traces for the major reactions. As well, there needs to be much more clarification and detail with respect to the reaction methodology. Please also address any formatting issues identified by the reviewers, particularly, please provide clearer images for Figure 1. I'm also in agreement with both reviewers that at times, the manuscript is difficult to read. Overall, the major points are very understandable, but some word choices or grammatical errors do make specific results harder to interpret. I would like to refer to Publication Guideline 5 from PLOS ONE:

"If the language of a paper is difficult to understand or includes many errors, we may recommend that authors seek independent editorial help before submitting a revision. These services can be found on the web using search terms like “scientific editing service” or “manuscript editing service.”

I recommend exploring these services so that the flow of the entire paper is more cohesive and the language a little more clear.

Reviewers' comments:

Reviewer's Responses to Questions

**Comments to the Author**

1. Is the manuscript technically sound, and do the data support the conclusions?

Reviewer #1: Partly

Reviewer #2: Yes

2. Has the statistical analysis been performed appropriately and rigorously? 

Reviewer #1: I Don't Know

Reviewer #2: Yes

3. Have the authors made all data underlying the findings in their manuscript fully available?

Reviewer #1: Yes

Reviewer #2: Yes

4. Is the manuscript presented in an intelligible fashion and written in standard English?

Reviewer #1: No

Reviewer #2: Yes

5. Review Comments to the Author

Reviewer #1: In this paper, Takaoko et al attempt to validate a computer-based methodology to predict the efficacy of mutant UGT1A1 binding and activity to potential substrates. The data suggest that they have produced a nice model. However, there are many deficiencies in terms of the method description and overall manuscript layout and verbage that made the paper difficult to interpret.

Major issues:

1. What was the normalizing control for transfection efficiency?

2. How were the glucuronide peaks confirmed for all substrates?

3. The authors should show representative LC-MS/MS runs for all assays.

4. What was the sensitivity of all of their assays? Minimal level of detection? Variability? How many times were each experiment run? Etc.

5. P. 10. Line 162. What SDM method was used?

6. P. 11. Line 174. What is the ‘UGT reaction mix’? This reviewer is not confident about the use of this procedure and it providing accurate results.

7. Lines 182-184. Why were PNPG and EE2 added to the reaction mix?

8. Too many supplemental tables and figures.

9. Fig. S1 is not well-explained.

10. Fig. S3 is not sufficiently clear.

11. Table S1 is not properly formatted.

Lots of English to correct, including the following:

P. 4. Line 51. Fix ‘UGT1A1s’. Change ‘especially’ to ‘including’.

Line 53. reported ‘on a’ binding model.

Lines 69-70. ‘sequence’ to ‘sequencing’.

Lines 76-77. Not great English.

Line 78. The word ‘concretely’ is not meaningful here.

Line 80. Change to ‘We subsequently developed…’.

P.5. Line96. Delete ‘by’.

P. 5. Line 99. Is solvate a word?

Line 111. Delete ‘We can see that’, and change ‘is’ to ‘was’.

P.6. Line 116. Change ‘ following’ to ‘docking’

Line 119. Add ‘The’ at the beginning of the sentence.

Do not put figure legends within the text of the manuscript – makes it difficult to read.

P. 10. Line 169. Delete the use of the term ‘entry clone’. Just say that the vector was used to generate a clone….

Line 166. Poor sentence.

P. 11. Line 171. ‘…used for UGT1A1 activity and luciferase assays’.

P. 13. Line 207. Use ‘We removed one of the UGT1A1 mutants…’

Line 215. Add ‘and’ L233R.

P. 14. Line 222. Finally, ‘a’…

Line 234. You are comparing the conjugation capacity of UGT1A1 against AAP and E2 with…

Table 1. The UDPGA rows are repetitive and unnecessary. The amino acid changes headers are also repetitive and unnecessary.

Table 3. Should be a supplemental table as it does not provide data.

Table 4. The same should be followed for Table 4 as described above for Table 1. For Table 4, why were no data provided for N400D and W461R – this should at least be explained in a footnote within the table.

P. 23. Line 348. You do not need this sentence.

Line 354. Poor sentence structure.

Line 359-60. Should be, ‘In our docking analysis using the AutoDock…’.

Lines 362, 374, 541, 543, 545. Should read, ‘UGT1A enzymes…’.

Line 364. Should read’ ‘including the number…’.

Lines 377-79. Don’t need this sentence.

Reviewer #2: The utility for this technology is exciting and results are encouraging. The introduction follows a logical line of reasoning for what the authors set out to accomplish and why it is important.

1. It would be helpful to define "molecular simulation” because it was challenging to determine whether it was synonymous with “in silico” throughout the paper.

2. Line 54, please add “amino acids” before listing them so the reader knows what the list is.

3. Line 98, please define TIP3P.

4. Line 103, please define MM.

5. I can’t see the Figure 1 images. Please enhance and possibly enlarge.

6. The location of the figure legends makes the text a little tougher to follow. It would be easier as a reviewer to have the figure legends at the end of the document with the figures. The first section of results is broken up by one figure legend and two tables, making it more work for the reader to figure out what the block of text is associated with.

7. There are several inconsistencies relating to the figures. For instance, when referenced within the text in parentheses, most times it appears as (fig X) and at least once, it appears as (fig. X). There are also spaces between figure numbers and panel letter in places and not in others. In the figure legends, some are listed as Panels “A,B” and other times as (A), (B).

8. It would be helpful for Table 1 and Figure 2 to be near each other so that they can be viewed together.

Figures 2 and 3 are showing the same data for acetaminophen and 17-B-estradiol in separate graphs and then in a single graph. The pattern is not repeated for the bilirubin and SN-38. I recommend the authors be consistent and either do both graphs for bilirubin and SN-38 as well or just show figure 3 for the acetaminophen and 17-B-estradiol.

9. It would be helpful for the authors to add a small figure legend within the graphic that says what the grey and black bars are for figures 3 & 4. Also, a small figure legend within figure 5 listing each of the identities of the symbols and the r value and slope.

10. I would like to see a citation for the sentence in lines 345-346. This may just be an oddly worded sentence since the text of the article is about mutations and polymorphisms that alter amino acids in the protein sequences; however, it should be either cited or altered.

11. It would be more impactful if the authors stated in the results section which correlations were statistically significant. It may be that all correlations mentioned in the results were statistically significant, but it still makes it more work for the reader to figure that out.

6. PLOS authors have the option to publish the peer review history of their article (what does this mean?). If published, this will include your full peer review and any attached files.

Reviewer #1: No

Reviewer #2: No

---

## [Author Response · Author response to Decision Letter 0]

14 Oct 2019

Please see 'Response to Reviewers'.

---

## [Editor Report · Decision Letter 1]

28 Oct 2019

PONE-D-19-22725R1

Establishment of the experimental procedure for prediction of conjugation capacity in mutant UGT1A1

PLOS ONE

Dear Dr. Takaoka,

Thank you for submitting your manuscript to PLOS ONE. After careful consideration, we feel that it has merit but does not fully meet PLOS ONE’s publication criteria as it currently stands. Therefore, we invite you to submit a revised version of the manuscript that addresses the points raised during the review process.

We would appreciate receiving your revised manuscript by Dec 12 2019 11:59PM. To enhance the reproducibility of your results, we recommend that if applicable you deposit your laboratory protocols in protocols.io, where a protocol can be assigned its own identifier (DOI) such that it can be cited independently in the future. For instructions see: http://journals.plos.org/plosone/s/submission-guidelines#loc-laboratory-protocols

We look forward to receiving your revised manuscript.

Kind regards,

Douglas Dluzen

Academic Editor

PLOS ONE

Additional Editor Comments (if provided):

Greetings Dr. Takaoka,

After careful review of your responses, I am submitting to you my decision that a minor revision is needed. I believe you have satisfactorily answered the reviewer's comments and provided the additional information required to address their concerns. I have included minor comments and clarifications needed after reading the revised manuscript incorporating your changes. Thank you for your patience and I hope you find my comments helpful. I look forward to seeing your response and final draft.

Editor's Comments:

Line 59: Please clarify the term 'newborn diseases'. Do you mean diseases in newborns? Or new diseases?

Line 66: Please clarify the UGT1A1*28 allele is the promoter.

Lines 72-74: Can you be more specific what is being conjugated for better clarify.

Lines 75-76: Can you be more specific what is being conjugated for better clarify.

Line 81: Please be more specific for what is being conjugated in this context.

Line 85: Are there previous models for this? If so, please provide references.

Line 163: Please define why this mutant was selected for Figure 1F. Is this just a represented graph or a different reason for looking at this particular variant.

Line 167: No paragraph indent needed.

Line 310: Please rewrite to clarify: :“S2 Table show the variables and constants of the mathematical model for the docking results for each UGT1A1 variant.” or similar so that it is more clear.

Lines 334-335: This should be clarified to "validated for each tested metabolite." And the paragraph indentation can be eliminated for the next sentence to create one paragraph.

Line 339-340: Figure 4 legend does not appear to have text outside of the title (assuming similar to text in figure 2 legend).

Line 348: There does not appear be text for Figure 6 outside of a title.

Line 360: Should be procedures, not procedure

Line 368: Should be clarified to say several categorized mutant UGT1A1s.

Line 385: Are these the only known UGT1A1 variants, or just the most studied? Please clarify. 

Table S1: Please add in guidelines to this table so that the values for the hydrogen bars and their mean A distance are easier to read/categorize for each category. 

Best,

Doug Dluzen

---

## [Author Response · Author response to Decision Letter 1]

30 Oct 2019

Please see the file of Response to Reviewers.

---

## [Editor Report · Decision Letter 2]

1 Nov 2019

Establishment of the experimental procedure for prediction of conjugation capacity in mutant UGT1A1

PONE-D-19-22725R2

Dear Dr. Takaoka,

We are pleased to inform you that your manuscript has been judged scientifically suitable for publication and will be formally accepted for publication once it complies with all outstanding technical requirements.

With kind regards,

Douglas Dluzen

Academic Editor

PLOS ONE
---

## [Editor Report · Acceptance letter]

8 Nov 2019

PONE-D-19-22725R2 

Establishment of the experimental procedure for prediction of conjugation capacity in mutant UGT1A1 

Dear Dr. Takaoka:

I am pleased to inform you that your manuscript has been deemed suitable for publication in PLOS ONE. Congratulations! Your manuscript is now with our production department. 

With kind regards,

on behalf of

Dr. Douglas Dluzen 

Academic Editor

PLOS ONE